# Design and Validation of the Index of Adherence to the Dietary Guidelines for Chile 2022 (GABAS-Index 17)

**DOI:** 10.3390/nu17223621

**Published:** 2025-11-20

**Authors:** Catalina Ramírez-Contreras, Jaime Crisosto-Alarcón, Solange Parra-Soto, Jorge Burdiles-Aguirre, Gianella Liabeuf, Lautaro Briones-Suárez

**Affiliations:** 1Departamento de Nutrición y Salud Pública, Facultad de Ciencias de la Salud y de los Alimentos, Universidad del Bío-Bío, Chillán 3780000, Chile; sparra@ubiobio.cl (S.P.-S.); gliabeuf@ubiobio.cl (G.L.); 2Departamento de Ciencias de la Rehabilitación en Salud, Facultad de Ciencias de la Salud y de los Alimentos, Universidad del Bío-Bío, Chillán 3780000, Chile; jcrisosto@ubiobio.cl (J.C.-A.); jburdiles@ubiobio.cl (J.B.-A.)

**Keywords:** validation, GABAS, dietary guidelines, dietary assessment, nutritional epidemiology

## Abstract

**Background/Objectives**: Adherence to national dietary guidelines is essential for promoting healthy eating and preventing chronic diseases. In Chile, the 2022 update introduced new evidence-based recommendations, but no validated tool is currently available to assess adherence. The aim of this study was to develop and validate a tool to assess adherence to the updated Chilean dietary guidelines. **Methods**: For this purpose, five expert judges evaluated the content validity using Aiken’s V (V ≥ 0.80). Reliability was assessed through a 21-day test–retest in 30 participants (≥18 years, mean age 38.9 years; 63.3% women) using the Intraclass Correlation Coefficient (ICC(3,1)), a two-way mixed-effects model to assess the absolute agreement of individual measurements, Standard Error of Measurement (SEM), and Minimal Detectable Change (MDC95) at the 95% confidence level. Internal consistency was assessed in 152 participants (≥18 years) examined via McDonald’s ω, and construct validity through confirmatory factor analysis (CFA) using the WLSMV estimator. **Results**: The GABAS-Index 17 showed high content validity (Aiken’s V = 0.93–1.00), good internal consistency (ω = 0.64–0.71), and accurate reliability (ICC = 0.905; SEM < 1; MDC95 = 2.1). Confirmatory factor analysis supported the proposed four-dimensional structure (CFI = 1.00; TLI = 1.02; RMSEA = 0.00), confirming strong factorial validity and internal coherence. **Conclusions**: These findings support the GABAS-Index 17 as an adequate and reliable tool for assessing adherence to the updated Chilean dietary guidelines. Although some psychometric aspects, such as the factorial structure, could be improved, the instrument performs well for its intended purpose of providing an overall adherence score. Its use can facilitate monitoring dietary patterns, support nutrition research, and inform public health strategies to improve diet quality in the Chilean population.

## 1. Introduction

The current assessment of dietary guidelines in Chile, as reflected in the recent literature, indicates persistent challenges in achieving optimal diet quality and adherence to recommended patterns across the population [1,2,3,4]. Multiple studies document that both adults and children in Chile exhibit low adherence to healthy and sustainable dietary patterns, with significant deficits in fruits, vegetables, and legumes, alongside excessive consumption of fats, sugars, red meats, and animal fats [2,5,6,7]. This challenge is not uniform, as spatial analyses reveal regional disparities [5]. In addition, recent evidence suggests the health risk associated with high consumption of ultra-processed foods. For instance, Monda et al. [8], in a recent narrative review, reaffirm the consistent epidemiological links between ultra-processed food intake and higher risk of obesity, metabolic dysfunction, and chronic diseases.

The Chilean guidelines and policy framework are aligned with international recommendations to limit the intake of added sugars, sodium, and saturated fats, while promoting minimally processed foods, whole grains, fruits, and vegetables [9,10]. There is also ongoing interest in promoting Mediterranean-like dietary patterns, which have demonstrated health benefits in the Chilean population, though current adherence remains low [7].

Several investigations have explored composite indices reflecting adherence to Food-Based Dietary Guidelines (FBDG) principles. The Healthy Eating Score (HES), derived from *Encuesta Nacional de Salud (ENS) 2016–2017* data, has demonstrated consistent inverse associations with depression [11] and all-cause mortality [12], indicating both construct and predictive validity. Earlier, Ratner et al. [13] proposed an index of global food quality based on Chilean guidelines, applied to a large university sample, though further psychometric validation is needed. In addition, the 2022 update of the Chilean FBDG introduces significant changes [14]; it is no longer just about *what* to eat but also *how* to eat, with *whom*, and *where* food comes from. The guidelines strongly encourage consuming fresh and minimally processed foods, promote mealtimes as shared, screen-free experiences, and, for the first time, include explicit principles of food sustainability, such as prioritizing local production, respecting seasonality, reducing waste, and caring for the planet. This approach is fully aligned with the Food and Agriculture Organization of the United Nations (FAO) and World Health Organization (WHO) guidelines on healthy and sustainable diets, which consider nutritional, social, cultural, and environmental dimensions [15,16].

The guidelines also promote daily habits that support physical and social well-being, such as cooking at home, sharing meals, and avoiding screens while eating, recognizing the value of eating as a cultural and relational practice. Evidence shows that shared family meals are associated with better diet quality, higher fruit and vegetable intake, and a lower risk of obesity in both children and adults [17], while eating while watching screens is related to higher consumption of ultra-processed foods and less attention to satiety signals [18,19]. Moreover, emerging biological research highlights how diet quality may influence obesity risk through mechanistic pathways. For example, recent findings by Moscatelli et al. [20] describe how dietary exposures and metabolic imbalances can affect bone marrow–derived stem cells involved in adipogenesis, providing biological plausibility for population-level associations between diet quality and obesity. These changes reflect evolving scientific evidence and the national epidemiological context, but they also create an urgent need for validated tools to assess adherence to the new recommendations.

Collectively, this evidence underscores the urgent need for standardized, validated tools to quantify adherence to the Chilean dietary guidelines and to monitor their evolution over time. Establishing and validating robust adherence indices tailored to the Chilean context will enable consistent tracking of dietary trends, facilitate international comparisons, and guide the design of targeted interventions to improve population-level nutrition and reduce chronic disease risk. Therefore, the present study aimed to develop and validate a tool to assess adherence to the updated Chilean dietary guidelines.

## 2. Materials and Methods

### 2.1. Instrument Characteristics

The GABAS-Index 17 is an instrument designed to assess adherence to the 2022 Chilean FBDG. The guidelines include ten key messages [14]: *1.**Eat fresh foods from fairs and established markets.* *2.**Add color and flavor to your day by including fruits and vegetables in every meal.* *3.**Eat legumes in stews and salads as often as possible.* *4.**Drink water several times a day and do not replace it with juices or soft drinks.* *5.**Consume dairy products at all stages of life.* *6.**Increase your intake of fish, seafood, and seaweed from authorized sources.* *7.**Avoid ultra-processed products and those with “HIGH IN” labels.* *8.**Share cooking duties and enjoy both new and traditional dishes.* *9.**Enjoy your meals at the table, eat with others when possible, and put away screens.* *10.**Protect the planet, save water, do not waste food, separate your trash, and recycle.*

Each item of the GABAS-Index 17 is dichotomously scored to record compliance or non-compliance with established dietary recommendations. This approach followed the methodology of international questionnaires that assess adherence to the Mediterranean diet [21,22]. The maximum possible score is 17 points. All items award 1 point for a “yes” response, except for item 14, which gives 1 point for a “no” response. Higher scores indicate greater adherence to the 2022 Chilean Dietary Guidelines.

In addition to the overall score, the index provides subscores across four dimensions: Food Safety (items 1 and 9), Healthy Eating (items 2, 3, 4, 5, 6, 7, 8, and 10), Eating Habits (items 11, 12, 13, and 14), and Food Sustainability (items 15, 16, and 17). The four dimensions were theoretically derived from key domains identified in current frameworks of healthy and sustainable eating behaviors. Food Safety refers to practices that promote the consumption of safe and fresh foods, emphasizing the purchase of fruits, vegetables, legumes, eggs, and fish from authorized markets with sanitary approval, in accordance with Chilean Dietary Guidelines that encourage eating fresh foods from fairs and established markets and choosing seafood from approved sources. Healthy Eating includes items that reflect the quality, balance, and nutritional adequacy of food intake, consistent with national dietary guidelines. Eating Habits address social and behavioral aspects of eating, such as sharing cooking duties, enjoying both traditional and new dishes, eating in company, and avoiding the use of screens during meals. These items reflect the Chilean Dietary Guidelines’ messages to share cooking responsibilities, enjoy diverse preparations, and foster mindful, social eating experiences. Finally, Food Sustainability involves environmentally responsible behaviors, such as avoiding food and water waste, separating waste, and recycling, aligned with the Chilean Dietary Guidelines’ message to protect the planet by saving water, reducing food waste, and recycling. The instrument is provided in Appendix A.

### 2.2. Content Validity

The content validity of the items was assessed using Aiken’s V coefficient, with the purpose of estimating the degree of agreement among expert judges regarding the relevance and quality of the criteria included in the scale. Five judges, all nutritionists with expertise in research, institutional food services, public health, epidemiology, clinical nutrition, and experience in instrument evaluation, participated in the process. Each judge rated the items according to eight criteria: sufficiency, pertinence, clarity, currency, objectivity, strategy, consistency, and structure, using a 4-point Likert scale, where 1 represented the lowest level and 4 the highest level of adequacy.

For each criterion, the Aiken’s V value was calculated, taking into account the number of judges and the range of the rating scale. The values were interpreted following methodological literature, assuming that V ≥ 0.80 indicates adequate content validity [23].

### 2.3. Reliability Assessment

For this purpose, a test–retest procedure was conducted with a 21-day interval between administrations according to Terwee et al. [24] in a sample of 30 participants aged 18 years or older (38.9 ± 10.0 years, range 24–74 years), 63.3% women, recruited by convenience sampling using a snowball methodology to enroll new participants.

Test–retest reliability for the GABAS-Index 17 was assessed using the Intraclass Correlation Coefficient (ICC) with a two-way mixed-effects model, absolute agreement, and single measures (which correspond to ICC(3,1)). This model was selected considering the intended future use of the instrument as a single-administration tool. The interpretation of ICC values followed Koo & Li [25]. Ninety-five percent confidence intervals (95% CI) were calculated using the F-distribution.

The ICC was computed for the total score of the GABAS-Index 17 as well as for each of its proposed subscales: Food Safety, Healthy Eating, Eating Habits, and Food Sustainability. In addition, to evaluate measurement precision and sensitivity to change, we computed the Standard Error of Measurement (SEM), the Minimal Detectable Change at the 95% confidence level (MDC95), and the Smallest Detectable Change proportion (SDC%). The SEM reflects the amount of random measurement error in observed scores, with lower values indicating greater precision. The MDC95 represents the smallest difference between two measurements that can be interpreted as a true change rather than random fluctuation. Thus, the MDC95 provides an estimate of the minimum score variation required to confidently conclude that a real change has occurred in the underlying construct. The SDC% expresses this threshold relative to the total possible score of the scale or subscale, providing an intuitive indicator of the proportion of the score range that must change to conclude that a real change has occurred.

### 2.4. Internal Consistency

For this procedure, the test was administered to 152 participants aged 18 years or older, recruited by convenience sampling, whose results were used in this section. The sociodemographic characteristics of the population are shown in Table 1. It is worth noting that this sample differs from the one used in the reliability assessment.

A RMSEA-based power analysis [26] conducted using the semTools package in R indicated that the sample size (*n* = 152) provided adequate statistical power (power = 0.90) to detect a not-close fit (RMSEA = 0.08) against the null hypothesis of close fit (RMSEA = 0.05), given the degrees of freedom of the model.

For each dimension or subscale, a congeneric model (free loadings) and a tau-equivalent model (equal loadings) were compared using confirmatory factor analysis (CFA) with the WLSMV estimator, which is appropriate for dichotomous items. In all cases with three or more items, the congeneric model showed a significantly better fit (*p* < 0.01), indicating that factor loadings differed across items. Therefore, the tau-equivalence assumption was rejected, and internal consistency was evaluated using McDonald’s omega (ω), computed separately for each dimension, which is appropriate in this context given the dichotomous nature of the items. For this reason, Cronbach’s alpha was not used, as it assumes tau-equivalence and can underestimate reliability under these conditions.

### 2.5. Construct Validity

A confirmatory factor analysis (CFA) was conducted to assess the factorial validity of the GABAS and to determine whether the hypothesized four-dimensional structure—Food Safety, Healthy Eating, Eating Habits, and Food Sustainability—fit the empirical data. As the questionnaire items were dichotomous (yes/no), the WLSMV (Weighted Least Squares Mean and Variance adjusted) estimator was employed, as recommended for categorical indicators. Model fit was evaluated using the Chi-square (χ^2^), Comparative Fit Index (CFI), Tucker–Lewis Index (TLI), Root Mean Square Error of Approximation (RMSEA), and Standardized Root Mean Square Residual (SRMR), considering CFI and TLI values ≥ 0.90 and RMSEA ≤ 0.08 as indicators of acceptable model fit. To ensure proper estimation and interpretation for binary items, CFA procedures followed contemporary methodological guidance for categorical data [27], acknowledging that certain indices—particularly SRMR—can behave differently in models with few items and limited response variability.

### 2.6. Ethical Considerations

All the study procedures were conducted according to the general recommendations of the Declaration of Helsinki and were approved by the Bioethics and Biosafety Committee of the University of Bio-Bio (CRC-08/2024).

## 3. Results

### 3.1. Content Validity Results

The results reveal a high level of agreement among judges regarding the content validity of the evaluated criteria (Table 2). The average ratings ranged between 3.8 and 4 points, reflecting an overall perception of high adequacy. The Aiken’s V coefficients were consistently high, ranging from 0.93 to 1.00, confirming the relevance and clarity of the assessed criteria.

In particular, the criteria Objectivity, Consistency, and Structure reached maximum values of V = 1.00, while the remaining ones (Sufficiency, Pertinence, Clarity, Currency, and Strategy) obtained values of V = 0.93, all within the excellent validity range. For all items, the lower bound of the 95% confidence interval remained between 0.80 and 1.00. These findings indicate that the judges considered the items to display high coherence, conceptual relevance, and linguistic clarity, providing strong evidence of content validity for the instrument and, in consequence, no revisions were deemed necessary following the expert review.

### 3.2. Reliability Assessment Results

According to the classification proposed by Koo and Li (2016) [25], the Food Safety dimension (ICC(3,1) = 0.57, 95% CI [0.270–0.770]) and the Healthy Eating dimension (ICC(3,1) = 0.735, 95% CI [0.514–0.864]) demonstrated moderate reliability. The relatively lower coefficient for Food Safety may be explained by the small number of items (two), which can statistically inflate response variability. The Eating Habits (ICC(3,1) = 0.892, 95% CI [0.786–0.947]) and Food Sustainability (ICC(3,1) = 0.877, 95% CI [0.758–0.939]) dimensions showed good reliability. For the overall GABAS-Index 17 scale, the instrument demonstrated excellent reliability (ICC(3,1) = 0.905, 95% CI [0.811–0.954]), although the ICC values for each scale show variable reliability, ranging from moderate to excellent.

Bland–Altman analyses (Figure 1) showed that all test–retest differences for the total GABAS-Index 17 score fell within the 95% limits of agreement, indicating excellent stability at the total-scale level. In contrast, the four subscales exhibited a small number of observations exceeding the limits of agreement, reflecting greater measurement variability in dimensions composed of fewer items. Nevertheless, the magnitude of these deviations was modest and did not indicate systematic bias across the score range. Overplotting was observed given the scale of the GABAS-Index 17 and its subscales.

All SEM values were below 1, indicating good measurement precision given the scale’s unit of measurement. This suggests that the standard error remained below the minimum possible score for a single item. The same pattern (SEM < 1) was observed for the total scale score. For the total test score, the MDC95 was 2.1. Considering that the maximum possible score is 17, this indicates good sensitivity to change, as a variation of approximately 12% is required to be considered a real change. Consistent with this, the SDC% for the total score was 36.14%. Across subscales, SDC% values ranged from 31.95% (Eating Habits) to 56.42% (Healthy Eating), reflecting varying degrees of detectable change relative to each subscale’s score range. Overall, the instrument demonstrated adequate precision and stability to detect true changes in the measured construct. These results support the temporal stability and measurement consistency of the GABAS-Index 17. These data are presented in Table 3.

### 3.3. Internal Consistency Results

For internal consistency analysis, McDonald’s omega (ω) was calculated based on tetrachoric correlations. The Healthy Eating (ω = 0.71) and Food Sustainability (ω = 0.71) dimensions showed acceptable levels of reliability, whereas Eating Habits (ω = 0.64) demonstrated moderate internal consistency. The Food Safety subscale did not allow the computation of ω due to the insufficient number of items. Overall, these results indicate that the instrument shows adequate internal consistency, although certain dimensions may benefit from further refinement to improve reliability. These findings provided the basis for subsequent analyses of the construct validity of the GABAS-Index 17.

### 3.4. Construct Validity Results

Before conducting the confirmatory factor analysis, an item-level analysis was performed to examine the individual behavior of each item. Table 4 reports endorsement rates, tetrachoric item–total correlations, and point-biserial item–total correlations for all 17 items of the GABAS-Index 17.

Item-level analyses indicated variability in endorsement rates across the 17 items, with some showing very high or very low prevalence (e.g., Items 1, 13, 15–16), which may limit their ability to differentiate between respondents. Tetrachoric and point-biserial item–total correlations were generally within the expected range for dichotomous items, with Items 2, 3, and 10 demonstrating the strongest relationships with the total score. A few items—such as Items 4, 9, and 17—showed weaker associations, and Item 11 exhibited minimal correlation with the total score. These item-level patterns are consistent with the variability in factor loadings observed in the CFA presented in the next section, supporting the interpretation of heterogeneous item functioning within the scale.

Then, the CFA was performed to confirm the four-dimensional structure (Food Safety, Healthy Eating, Eating Habits, and Food Sustainability) derived from the conceptual framework underlying the instrument. This analysis, estimated using the DWLS/WLSMV method for ordinal variables, showed an adequate overall model fit. Goodness-of-fit indices indicated a strong correspondence between the four-dimensional theoretical model and the observed data (χ^2^ (113) = 108.61, *p* = 0.60; CFI = 1.00; TLI = 1.02; RMSEA = 0.00 [90% CI 0.00–0.04]; SRMR = 0.12). These results suggest that the proposed four-factor structure—Food Safety, Healthy Eating, Eating Habits, and Food Sustainability—is consistent with participants’ responses. Most standardized factor loadings were moderate to high (ranging from 0.30 to 0.95, except for the items in the Food Safety dimension), with particularly strong associations for items within the Healthy Eating and Food Sustainability factors, supporting the internal coherence of these dimensions (Table 5). However, a few items exhibited low or non-significant loadings, suggesting the need to review their wording or conceptual alignment with the construct. Overall, these findings provide preliminary evidence of factorial validity for the GABAS-Index 17, indicating that the four-factor model fits the data well, although minor item-level refinements may further strengthen its psychometric performance. It should be noted that, for the computation of standardized factor loadings, the model was estimated by fixing the variance of the latent factors to 1, in order to obtain a stable scaling of the factors. Modification indices (MI) were examined, and none exceeded the recommended threshold of 10, indicating that no substantial residual correlations or cross-loadings were suggested by the model. Therefore, no additional model modifications were implemented.

## 4. Discussion

This study evaluated the psychometric properties of the GABAS-Index 17, which was designed to measure adherence to the 2022 Chilean Dietary Guidelines. The instrument demonstrated excellent temporal reliability (ICC = 0.905) and good absolute precision (SEM < 1; MDC95 = 2.1), as well as acceptable internal consistency (ω = 0.64–0.71). The confirmatory factor analysis supported the hypothesized four-dimensional structure—Food Safety, Healthy Eating, Eating Habits, and Food Sustainability—showing excellent model fit (χ^2^ (113) = 108.61, *p* = 0.60; CFI = 1.00; TLI = 1.02; RMSEA = 0.00; SRMR = 0.12). Most standardized loadings were moderate to high, indicating that the items adequately represent their respective latent constructs.

Unlike other dietary quality indices used in the region [28,29], the GABAS-Index 17 incorporates specific components aligned with the 2022 Chilean Food-Based Dietary Guidelines [14]. These features support its validity and reliability as an instrument for measuring adherence to the updated guidelines, facilitating the monitoring of dietary patterns, advancing nutrition research, and informing public health strategies aimed at improving population diet quality.

The need for precise adherence instruments is highlighted by global dietary patterns. A systematic review found that approximately 40% of populations worldwide do not meet their national dietary guidelines, with adherence to fruit and vegetable recommendations ranging from 7% to 67.3% [30]. While international tools such as the Healthy Eating Index (HEI) assess general dietary patterns, the GABAS-Index 17 is specifically tailored for Chile. It is the only instrument designed to measure adherence to the progressive 2022 Chilean guidelines, including dimensions such as food sustainability and avoidance of ultra-processed foods, which are typically absent from other national indices.

A growing body of evidence shows that high consumption of ultra-processed foods (UPF) is closely linked to increased obesity risk, weight gain, and unfavorable metabolic outcomes [8]. This evidence offers strong conceptual support for incorporating an “avoid ultra-processed foods” domain within dietary adherence indices. In the Chilean context, this dimension is particularly relevant, as it aligns with ongoing regulatory efforts (front-of-package warning labels and restrictions on food marketing) [31,32]. Thus, integrating this component into the GABAS-Index 17 enhances its policy relevance and strengthens its ability to monitor behaviors associated with key determinants of inadequate diet quality in Chile.

In this context, recent research on healthy and sustainable diets highlights that incorporating conscious eating practices, such as preferring fresh and locally sourced foods, reducing waste, and choosing homemade preparations, is associated with better eating patterns and greater overall well-being [15,33]. However, despite this evidence, adherence rates remain concerning. Although dietary guidelines are a fundamental public health tool, their true impact depends on each country’s ability to assess and monitor compliance in a contextualized manner, considering cultural, economic, and environmental factors that influence dietary habits.

It is important to note that indices such as the HEI-2015, the Alternative Healthy Eating Index (AHEI) [34,35], and the Mediterranean Diet Adherence Screener (MEDAS-14) [36] typically report test–retest reliability coefficients ranging from 0.70 to 0.90, values comparable to those observed in the present study (total ICC = 0.905). In Chile, previous instruments such as the HES derived from the 2016–2017 ENS [11,12] demonstrated predictive validity for health outcomes such as depression and mortality but lacked formal psychometric analyses. Similarly, the Index of global food quality did not report reliability or internal validity estimates [13]. In this context, the GABAS-Index 17 represents a relevant methodological contribution, incorporating comprehensive statistical evidence on stability, precision, and internal consistency, and being explicitly aligned with the 2022 Food-Based Dietary Guidelines update, which introduces the axes of sustainability and food security.

From a methodological perspective, the results demonstrate excellent temporal stability of the instrument as a whole, with an intraclass correlation coefficient (ICC = 0.905) indicating high consistency of responses over time in the absence of interventions. This stability, along with adequate absolute precision values (SEM < 1; MDC95 = 2.1), shows that the GABAS-Index 17 can detect real changes of moderate magnitude (≈12%), making it a sensitive tool for monitoring nutritional interventions. The subscales showed reliability levels consistent with their length: Food Safety showed moderate reliability (ICC = 0.57), as expected given its brevity (two items), while Eating Habits (ICC = 0.89) and Food Sustainability (ICC = 0.88) achieved high values, indicating adequate stability across diverse behavioral dimensions [25]. For internal consistency, McDonald’s ω coefficients ranged from 0.64 to 0.71, values considered acceptable for scales with dichotomous items and multidimensional constructs [37]; however, certain dimensions could benefit from additional improvement to strengthen their reliability. Additionally, the CFA results moderately support the proposed four-dimensional theoretical structure—Food Safety, Healthy Eating, Eating Habits, and Food Sustainability—with good fit indices (χ^2^ (113) = 108.61, *p* = 0.60; CFI = 1.00; TLI = 1.02; RMSEA = 0.00; SRMR = 0.12). These values exceed the commonly accepted thresholds for a good-fitting model (CFI/TLI ≥ 0.90; RMSEA ≤ 0.08) [38], confirming the factorial validity of the instrument and the internal consistency of its dimensions. The moderate to high factor loadings show a good association between items and factors, especially in the Healthy Eating and Food Sustainability dimensions, as these dimensions contain more items, whereas the remaining factors exhibit only moderate loadings, likely attributable to the limited number of items. Although the sample size is modest, RMSEA-based power analysis showed that statistical power was sufficient (90%) for evaluating model fit. Nevertheless, replication with larger samples is recommended.

It is worth noting that the GABAS-Index 17 relies on equal item weighting, a design choice rather than an empirically optimized scoring method. Given the heterogeneity of item loadings, future work may examine whether differential item weights based on factor loadings or item information enhance scoring accuracy. These findings indicate that the GABAS-Index 17 is a robust, stable, and conceptually coherent instrument that distinctly captures the key behavioral domains promoted by the 2022 FBDG.

From a biological standpoint, diet quality influences adiposity and metabolic health through several interconnected mechanisms. Diets high in saturated fats, refined sugars, and ultra-processed foods promote chronic low-grade inflammation, impair insulin sensitivity, and disrupt appetite-regulation pathways involving leptin, ghrelin, and incretins [39]. Emerging evidence suggests that unhealthy dietary exposures may affect the differentiation and function of mesenchymal stem cells, favoring adipogenic pathways and reducing metabolic flexibility [20,40]. In contrast, diets rich in fiber, polyphenols, unsaturated fats, and minimally processed foods support mitochondrial function, reduce oxidative stress, and improve insulin action [41]. Together, these mechanisms underscore the clinical relevance of monitoring adherence to the behaviors captured by the GABAS-Index 17, as they reflect dietary exposures with direct implications for obesity prevention and metabolic health.

This study has some limitations. Because the sample was highly unbalanced in terms of sex (97% women) and showed limited age variability (M = 48.8, SD = 10.8), multi-group CFA models could not be reliably estimated. As a result, preliminary tests of measurement invariance across sex and age bands were not feasible. Future studies with more balanced and age-diverse samples should assess the instrument’s invariance. Some subscales, such as Food Safety, include a small number of items, which may restrict statistical stability and the estimation of internal reliability. The sample size of the test–retest analysis (*n* = 30) could limit the precision of the confidence intervals, and the overall sample (*n* = 152) does not guarantee national representativeness, so it is recommended to replicate the validation in larger and more diverse groups. Another potential limitation is the use of dichotomous (“yes/no”) responses, which, although consistent with the structure of the Chilean Food-Based Dietary Guidelines and other adherence questionnaires, may reduce the instrument’s sensitivity to detect gradations in dietary behaviors. Future adaptations of the instrument could explore ordinal scoring schemes to enhance discrimination without compromising simplicity and public health applicability. Nonetheless, the study has strengths that should be considered. First, it is the pioneering instrument specifically designed and validated to assess adherence to the 2022 Chilean Dietary Guidelines, explicitly integrating the principles of Sustainability and Food Safety. Second, the psychometric validation was comprehensive and included analyses of temporal reliability, absolute precision, internal consistency, and factor structure, using statistical methods appropriate for dichotomous data (ICC(3,1) model, tetrachoric correlations, and WLSMV estimator). This approach provides a solid basis for the use of the GABAS-Index 17 in population-based research and food policy evaluation programs. Future research should focus on further validating the GABAS-Index 17 by assessing its convergent and discriminant validity against other dietary quality indicators, such as the HEI, the AHEI, and the HES, which have been used in recent Chilean studies. It is also important to examine its relationship with nutritional and metabolic health biomarkers to determine its criterion validity and its ability to predict relevant clinical outcomes. Additionally, future research should consider adapting and validating the index in different population groups, such as children and adolescents.

## 5. Conclusions

In summary, our findings indicate that the GABAS-Index 17 is a valid and reliable instrument for assessing adherence to the updated Chilean Dietary Guidelines. Although the factorial structure could be further improved, this is not critical to its intended use, as the instrument is designed to provide an overall adherence score rather than independent subscale scores. The organization of items into four dimensions reflects an operational grouping based on predefined guideline recommendations rather than a strictly empirical latent structure. Despite some limitations, such as the modest internal consistency of certain subscales, the limited sample size, and the absence of criterion validation, the GABAS-Index 17 demonstrates robust psychometric properties, with excellent test–retest reliability (ICC = 0.905) and acceptable internal consistency (ω up to 0.71). These results support its use in research and public health monitoring to evaluate dietary behaviors and promote healthier eating patterns in Chile. Future research should continue to strengthen the validity of the GABAS-Index 17 by examining its convergent performance relative to other dietary quality indices, evaluating criterion validity against adiposity and metabolic biomarkers, and assessing measurement invariance across different demographic groups.

## Figures and Tables

**Figure 1 nutrients-17-03621-f001:**
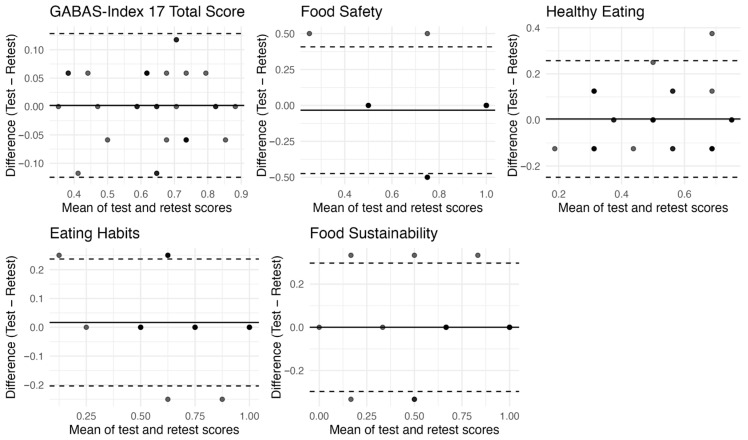
Bland–Altman plots for the total score and subscale scores of the GABAS-Index 17. The plot shows the mean difference (solid line) and the limits of agreement (dashed lines; mean ± 1.96 SD).

**Table 1 nutrients-17-03621-t001:** Characteristics of the population studied.

	*n* = 152
Women, % (*n*)	97.4 (148)
Age, years	48.8 (10.8)
Education	
Primary, % (*n*)	7.9 (12)
Secondary % (*n*)	49.3 (75)
University, % (*n*)	42.8 (65)
Marital status	
Single, % (*n*)	23.7 (36)
Married, % (*n*)	44.7 (68)
In a relationship, % (*n*)	19.7 (30)
Separated/Divorced, % (*n*)	11.8 (18)

Values are means (standard deviations) for continuous data and percentages % and absolute values (*n*) for categorical data.

**Table 2 nutrients-17-03621-t002:** Content validity rating matrix.

Dimension	Judge 1	Judge 2	Judge 3	Judge 4	Judge 5
Sufficiency	4	4	4	3	4
Pertinence	4	4	4	3	4
Clarity	4	3	4	4	4
Currency	4	4	4	3	4
Objectivity	4	4	4	4	4
Strategy	4	4	4	3	4
Consistency	4	4	4	4	4
Structure	4	4	4	4	4

Meet the criteria = 4; Moderate level = 3; Low level = 2; Does not meet criterion = 1.

**Table 3 nutrients-17-03621-t003:** Reliability analysis results for the GABAS-Index 17 and its dimensions.

Dimension	ICC(3,1)	95% CI Lower Limit	95% CI Upper Limit	SEM	MDC95	SDC%
Food Safety	0.571 ***	0.27	0.77	0.153	0.423	49.27%
Healthy Eating	0.735 ***	0.514	0.864	0.092	0.256	56.42%
Eating Habits	0.892 ***	0.786	0.947	0.074	0.205	31.95%
Food Sustainability	0.877 ***	0.758	0.939	0.109	0.303	43.18%
GABAS-Index 17 Total	0.905 ***	0.811	0.954	0.076	0.213	36.14%

Note. *** = *p* < 0.001; ICC = Intraclass Correlation Coefficient; CI = Confidence Interval; SEM = Standard Error of Measurement; MDC95 = Minimal Detectable Change at 95% confidence level; SDC% = Smallest Detectable Change proportion.

**Table 4 nutrients-17-03621-t004:** Endorsement Rates and Item–Total Correlations for the GABAS-Index 17 Items.

Item	Endorsement Rate	Tetra Item Total	Point Biserial Item Total
Item 1	0.941	0.299	0.15
Item 9	0.382	0.463	0.363
Item 2	0.428	0.409	0.324
Item 3	0.592	0.167	0.132
Item 4	0.671	0.292	0.225
Item 5	0.401	0.244	0.192
Item 6	0.237	0.336	0.244
Item 7	0.309	0.293	0.223
Item 8	0.776	0.177	0.127
Item 10	0.612	0.484	0.381
Item 11	0.474	0.004	0.004
Item 12	0.783	0.333	0.238
Item 13	0.901	0.341	0.199
Item 14	0.408	0.263	0.208
Item 15	0.888	0.207	0.125
Item 16	0.928	0.205	0.109
Item 17	0.289	0.176	0.133

Note: Endorsement Rate = “Yes” prevalence; Tetra Item Total = Tetrachoric item–total correlation; Point Biserial Item Total = Point-Biserial Item Correlation.

**Table 5 nutrients-17-03621-t005:** Standardized Factor Loadings and Residual Variances for the CFA Model.

Factor	Item	Std. Loading	Residual Variance
Food Safety	Item 1	0.016	1.000
	Item 9	0.005	1.000
Healthy Eating	Item 2	0.733	0.462
	Item 3	0.581	0.663
	Item 4	0.179	0.968
	Item 5	0.518	0.732
	Item 6	0.428	0.817
	Item 7	0.424	0.820
	Item 8	0.308	0.905
	Item 10	0.724	0.475
Eating Habits	Item 11	0.131	0.983
	Item 12	0.646	0.583
	Item 13	0.734	0.461
	Item 14	0.430	0.815
Food Sustainability	Item 15	0.813	0.339
	Item 16	0.949	0.098
	Item 17	0.008	1.000

Note. Std. Loading = standardized factor loading.

## Data Availability

The original data presented in the study are openly available in OSF at https://doi.org/10.17605/OSF.IO/YKQ2J.

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
