# Peer review of "Design and Validation of the Index of Adherence to the Dietary Guidelines for Chile 2022 (GABAS-Index 17)"

_nutrients, 2025, doi:10.3390/nu17223621_

Round 1

Reviewer 1 Report

Comments and Suggestions for Authors

The manuscript addresses an important public health issue — the lack of validated instruments to assess adherence to Chile’s updated 2022 Food-Based Dietary Guidelines (FBDG). Developing such a culturally specific, psychometrically sound index is valuable. However, while the topic is relevant and the structure generally follows standard validation procedures, the manuscript suffers from serious methodological weaknesses, unclear reporting, redundancy, and limited interpretability of results. The statistical analyses are superficially presented, and the sample characteristics and justification for statistical choices are insufficiently described. Overall, the paper would require substantial methodological and reporting improvements .

  • The sample size used for validation (n = 152) and for test–retest reliability (n = 30) is too small to support robust confirmatory factor analysis (CFA) and generalizable reliability estimates.
    → Suggestion: Authors should provide a power analysis or theoretical justification for these numbers and acknowledge more explicitly the limitations of such small samples in factor modeling.
  • No demographic or socioeconomic description of the participants is provided (age, sex distribution, educational level, region, etc.), which precludes assessing sample representativeness or possible biases.
  • The recruitment method (convenience sampling? online? clinical?) is not described clearly, which affects the external validity of the findings.
  • The process of item selection from the FBDG messages is not transparent. It appears that the authors created one dichotomous question per guideline but did not empirically test whether each item adequately captures the intended construct.
    → Suggestion: Explain how items were operationalized, whether cognitive interviews or pilot testing were conducted, and if any items were revised based on participant feedback.
  • The decision to use dichotomous (“yes/no”) responses may oversimplify complex behaviors like food sustainability or meal sharing. Such binary coding can reduce sensitivity and reliability. Authors should justify why Likert-type scaling was not considered.
  • Content validity: Aiken’s V = 0.93–1.00 suggests perfect expert agreement, which is unrealistic given five judges and eight criteria. Authors must provide the actual rating matrix or at least range/variance to support these claims.
  • Reliability: Although ICC = 0.905 for total score is high, ICCs for subscales (0.57–0.73) are only moderate. The authors interpret all values as “good” or “excellent,” which is misleading. In addition, SEM and MDC95 are reported but not clearly explained in context.
  • Internal consistency: McDonald’s ω values (0.64–0.71) are marginal and would not be considered “good.” The discussion glosses over this weakness.
  • Construct validity: CFA results are implausibly perfect (CFI = 1.00; TLI = 1.02; RMSEA = 0.00). With small samples and dichotomous data, such fit indices likely indicate overfitting or estimation artifacts. The authors must report standardized loadings, residuals, and model modification indices to allow evaluation.
  • The discussion is overly self-congratulatory, restating positive results without critical analysis. Limitations are only superficially mentioned.
    → Suggestion: Expand the limitations section substantially, acknowledging sample bias, small sample size, potential response bias from dichotomous items, and the absence of external validation (e.g., correlation with other dietary indices or biomarkers).
  • The conclusion that the index is “valid and reliable” is overstated given the modest reliability, limited sample, and absence of criterion validation.
  • The text is redundant (e.g., repeated phrases like “adequate precision and stability” appear twice).
  • English is mostly understandable but awkward and repetitive, suggesting heavy use of automated language editing. The flow of the introduction could be improved by reducing background detail and clarifying the study’s rationale and novelty.
  • Tables and Appendix A could be better formatted for readability.

Reviewer 2 Report

Comments and Suggestions for Authors

The manuscript entitled ““ presents the development and psychometric validation of the GABAS-Index 17, an innovative tool for assessing adherence to the Chilean Food-Based Dietary Guidelines, updated in 2022. The topic is highly relevant for nutritional epidemiology and public health surveillance in Latin America. The paper is well organized, methodologically sound, and clearly written. With a few minor corrections I recommend for publication.
The manuscript does not provide any demographic description (age, sex, education, socioeconomic level) of participants in any of the samples. Please include a table describing the sample characteristics and recruitment methods. This is essential to assess external validity.
The use of dichotomous responses simplifies scoring, but reduces sensitivity. Please discuss this aspect in the discussion.
Explicitly justify the use of ω instead of Cronbach’s α (a good choice here), but mention that these coefficients could improve with multiple items per dimension.
Ethics approval, consent, and transparency are adequately addressed. No comments.
Abstract: First sentence is long and dense. Simplify for clarity.
Methods: Use consistent subheadings (e.g., “2.2 Reliability Assessment” should be “2.3,” etc.). Check numbering and structure.
In the results section, remove duplicate sentences for brevity.
Overall, the manuscript is well written.

Reviewer 3 Report

Comments and Suggestions for Authors

The abstract is informative but currently leans on superlatives that are difficult to justify in light of a few borderline or mixed indices later in the text. I suggest softening formulations such as “excellent content validity,” “excellent reliability,” and “strong factorial validity,” and instead briefly presenting key numbers with context. For example, report the expert panel size for Aiken’s V, the exact ICC model and 95% CI, the ω range with its interpretation for dichotomous items, and at least one distributional index (e.g., floor/ceiling) for total scores. Stating the test–retest interval (21 days), the two analytic samples (n=30; n=152), and basic participant descriptors in the abstract will increase transparency and strengthen the take-home message.

The introduction provides a thorough national and international framing and argues convincingly why adherence tools must now reflect sustainability, screen-free shared meals, and reduced ultra-processed food consumption—axes explicitly present in the 2022 FBDG. Given this emphasis, the background would benefit from integrating two recent strands of literature that directly reinforce the rationale for specific items in the GABAS-Index 17. First, the narrative synthesis on ultra-processed foods and obesity risk (Monda et al., 2024, Foods) would complement the guideline’s “avoid ultra-processed” message and situate item 10 within a contemporary epidemiologic consensus. Second, work illuminating biological underpinnings of obesity involving bone marrow stem cells (Moscatelli et al., 2024, IJMS) can reinforce the introduction’s rationale for monitoring diet quality as a pathway to obesity prevention, lending mechanistic depth to the public-health framing. These additions are not strictly necessary for the psychometric argument but will strengthen the manuscript’s translational arc from dietary advice to health outcomes.

Methods are generally rigorous and clearly explained, with appropriate choices for dichotomous data. Content validity via Aiken’s V is well chosen; please add item-level V estimates with confidence intervals and clarify the decision rule used (e.g., V≥0.80 as acceptance threshold) alongside any item revisions made after the expert round. Specify the experts’ disciplinary mix to assure content coverage across nutrition epidemiology, psychometrics, and public health. For reliability, the ICC specification (two-way mixed, absolute agreement, single measures) is appropriate for a one-administration tool, and the 21-day interval is defensible; please add a brief justification for this interval vis-à-vis potential memory effects in self-report behavioral items. It would also help to present Bland–Altman plots for total scores and subscale scores to visualize agreement and any heteroscedasticity across the score range, and to accompany SEM and MDC95 with an intuitive expression relative to the 0–17 scale (you note ≈12% already; keep this in the main text and the abstract). Consider adding the Smallest Detectable Change proportion (SDC%) and, if feasible, the Standard Error of Measurement at the subscale level to parallel the total-scale indices.

Internal consistency is handled with McDonald’s ω based on tetrachoric correlations, which is commendable. Because your dimensions are short and heterogeneous by design, ω in the 0.64–0.71 range is acceptable, but readers will appreciate two clarifications. First, report confidence intervals for ω; second, briefly justify your rejection of tau-equivalence with the CFA comparison you already conducted, and make explicit that α was not used because its assumptions are violated in this context. An item-level table with tetrachoric item–total correlations, endorsement rates (“Yes” prevalence), and point-biserial correlations would be very helpful, especially given that a few items later show low or non-significant factor loadings.

The CFA section is a strong point, yet some details merit attention. Fit indices are remarkably high (CFI=1.00; TLI=1.02; RMSEA=0.00; χ² non-significant), while SRMR=0.12 is above conventional cutoffs, which introduces a tension that deserves comment. Please discuss this divergence explicitly; in short measures with categorical indicators and modest sample sizes, SRMR can be inflated, but readers should not be left to reconcile this themselves. Provide the item-factor loadings with standard errors and significance, factor correlations with CIs, and a note on any residual correlations allowed (ideally none). A short paragraph on model diagnostics—modification indices, local dependence checks, and an a priori decision not to respecify—would enhance transparency. If space permits, consider reporting an exploratory bifactor or ESEM sensitivity analysis to show that the four-factor solution is preferable to a general-factor alternative given the index’s multidimensional intent. Finally, since the tool is intended for broad population use, a preliminary, table-based test of measurement invariance across sex and age bands (configural/threshold invariance at minimum) would strengthen construct validity claims; non-invariance, if present, can be flagged for future work.

Scoring and content deserve a brief methodological refinement. The binary scoring is pragmatic and keeps the instrument short, but several items—water intake, dairy portions, fish frequency—are intrinsically ordinal. Please comment on why you chose dichotomization over ordered categories and, if data permit, add a sensitivity analysis comparing the current binary scoring with a simple ordinal scheme (e.g., 0–1–2) using graded-response IRT or polychoric-based reliability to show whether discrimination improves without undermining feasibility. This is particularly relevant for subscales where ω is modest and for Food Safety, where the two-item length mechanically limits reliability. You might also consider, in the limitations, that equal weighting of items is a design choice rather than an empirical result; a short note acknowledging that future work could explore differential weights based on factor loadings or item information would be appropriate.

Results are clearly presented. Table 1 is useful and well formatted; adding a companion table with CFA loadings and item endorsement will make the section fully self-contained. Where you duplicate interpretive sentences verbatim (“Overall, the instrument demonstrated adequate precision and stability…” appears twice), please edit for concision. The discussion accurately situates findings among prior Chilean indices and international tools and properly emphasizes the novelty of embedding sustainability and screen-free, shared meals within an adherence index. Two targeted additions would sharpen this section. First, link your “avoid ultra-processed foods” dimension to the up-to-date synthesis on UPF and obesity mentioned above; this both justifies the content domain and underscores policy relevance in Chile’s regulatory landscape. Second, when you discuss the public-health implications for obesity, add one paragraph articulating plausible biological pathways whereby diet quality relates to adiposity and metabolic health—this is where a brief citation of the stem-cell literature fits and helps anchor the instrument’s clinical salience beyond behavioral measurement.

The conclusions are appropriately measured, though they could more explicitly reflect the limits you already acknowledge: small test–retest sample, subscale length constraints, and lack of external validity against biomarkers or hard outcomes. A forward-looking sentence specifying the next validation steps—convergent validity with HEI/AHEI/HES in a larger Chilean sample, criterion validity against adiposity or metabolic markers, and invariance testing—will make the closing paragraph more actionable.

Presentation and style require only minor polishing. Ensure consistent terminology for organizations (use “WHO” throughout instead of mixing with the Spanish “OMS”), check all abbreviations on first use, and harmonize tense between sections. The appendix is a valuable asset; consider providing an English translation of the items in Supplementary Material to maximize international uptake, alongside scoring instructions, a worked example, and ready-to-use syntax (R/lavaan or Mplus) for CFA and reliability calculations. The ethics, data availability, and authorship statements are standard; if possible, deposit de-identified data sufficient to reproduce the psychometric analyses, or at least the item-level response matrix, in a public repository to facilitate secondary validation.

Finally, the reference list is solid and appropriately international. I recommend adding the two 2024 references noted above to strengthen the conceptual bridge between dietary patterns—particularly the avoidance of ultra-processed foods—and obesity-related outcomes and mechanisms. In addition, when citing fit-index conventions and ω interpretation, consider adding a brief pointer to contemporary guidance on categorical CFA and reliability for binary indicators if space allows, to help readers unfamiliar with these nuances.

Monda el al., Ultra-Processed Food Intake and Increased Risk of Obesity: A Narrative Review, Foods. 2024 Aug 21;13(16):2627. doi: 10.3390/foods13162627; Moscatelli et al., Exploring the Interplay between Bone Marrow Stem Cells and Obesity, Int J Mol Sci. 2024 Feb 27;25(5):2715. doi: 10.3390/ijms25052715.

In sum, this manuscript fills an important methodological gap with an instrument that is brief, policy-aligned, and psychometrically promising. Addressing the points above—clarifying a few fit inconsistencies, expanding item-level reporting, modestly tempering the tone in the abstract, and enriching the background with the latest literature on ultra-processed foods and obesity mechanisms—should be feasible without additional data collection and will, I believe, position the GABAS-Index 17 as a credible, citable tool for research and surveillance in Chile and potentially as a model for other settings. My recommendation is minor revision.

Round 2

Reviewer 1 Report

Comments and Suggestions for Authors

 Accept in present form